# Determinants of Refugee Children's Social Integration: Evidence from Lebanon, Turkey, and Australia

**Mohammad Hammoud** [1,2,*], **Maha Shuayb** [1,3] **and Maurice Crul** [1]

1  Sociology Department, Vrije Universiteit Amsterdam, De Boelelaan, 1081 HV Amesterdam, The Netherlands
2  Centre for Lebanese Studies, Lebanese American University, Beirut 13-5053, Lebanon
3  Education Department, University of Cambridge, Cambridge CB2 8PQ, UK
*  Correspondence: mohammad.hammoud@lebanesestudies.com

**Abstract:** This paper investigates the determinants of refugee students' social integration in Lebanon, Turkey, and Australia. This paper seeks to understand how legal status and the corresponding length of refugee asylum shape refugee children's social integration. The three host countries offer refugees different legal statuses ranging from short-term in Lebanon, medium-term in Turkey, and long-term in Australia. Therefore, our data collection covers a sample of 1298 middle school refugee students from all three countries. Our probit regression analysis sheds light on the importance of micro-level factors related to individual and household characteristics and meso-level factors related to school factors shaping refugee students' social integration. The statistical dominance of meso-level factors indicates that the within-country differences are stronger than the between-country differences, yet it does not rule out the importance of macro policies that indirectly influence refugee students' social integration by shaping provisions at the micro and meso levels.

**Keywords:** refugee students; social integration; education in emergency; provisions; legal status

## 1. Introduction

Following the Syrian crisis that erupted into civil war in early 2011 and the subsequent massive influx of Syrian refugees to neighbouring countries, refugee research has proliferated considerably in the past decade (Shuayb and Crul 2020). With most research concerned about the immediate humanitarian response, studies investigating refugees' long-term education and resettlement goals are still scarce. A long-term vision would necessarily include a fully integrated refugee population within a host country's borders. Such a long-term process might be more successful for younger individuals, who can be integrated educationally, socially, and culturally into the host society if given the right opportunity. However, the resettlement process poses great challenges for refugee children, such as navigating a new schooling environment, family separation, cultural dissonance, difficulties learning a foreign language, acculturation stress, and limited financial resources (Roy and Roxas 2011). These factors combined often hinder the smooth resettlement process of refugee students and their ability to access and master certain social life areas (Laurentsyeva and Venturini 2017).

Social integration is a complex concept that does not fit within a single definition (Council of Europe 1997). However, the literature considers social integration as a multidimensional process that starts upon arrival in the host country (Castles et al. 2002). Crul et al. (2016) argue that different integration domains, such as education, employment, culture, and citizenship, are interdependent and that policies implemented in one domain will affect other domains. Their finding can be related to some critical evidence revealing the importance of educational integration with the employment system (Münz 2017), the role of language proficiency in both education and work (Crul et al. 2016), and the importance of clarifying legal status so that integration is quicker (Zimmermann 2016).

Educational integration is a good starting point for integrating refugee children into the host community. However, educational integration is pointless in the absence of prospects for economic, social, cultural, and political integration. Institutional contexts vary greatly from one country to another, and different institutional contexts produce distinct integration arrangements for refugees in education, labour market participation, access to housing, and legislation. Having said that, the comparative integration theory argues that social integration and belonging to the host country's community are dependent on the integration context prevailing within the host country (Crul and Schneider 2010). Therefore, the comparative integration theory suggests that we shift our focus from why individuals fail to socially integrate to why institutions lack inclusivity. Crul and Schneider (2010) argue that integration practices are pre-defined by certain institutional contexts, such as legal status, access to citizenship, and policies that shape channels of inclusion in education and employment. The comparative integration theory provides a solid baseline for the following comparison of the two education paradigms, which are also dependent on institutional contexts shaped by the type of legal settlement. "Emergency in education" and "long-term" are two education paradigms adopted by host countries following the Syrian crisis in 2011.

## 2. Education Paradigms

Education in Emergency (EIE) is mainly adopted by countries in the global south, countries that share borders with refugees' home country, and countries hosting refugees temporarily. In contrast, the long-term education paradigm is mostly adopted in the global north and is a product of the permanent residency the host country offers to its refugees (Shuayb and Crul 2020; Brun and Shuayb 2020). Each education paradigm is implemented using different policies, thus exposing refugee children to different schooling systems, integration provisions, languages, and curriculums. EIE is based on a human rights philosophy; its most common framework was established by the Inter-Agency Network for Education in Emergencies (INEE) and was adopted in 11 countries over the past ten years. While education is a long-term process that aims at preparing children for the future, humanitarianism is concerned with the immediate needs of people affected by emergencies. The conflicting aims between education and humanitarianism produce different educational challenges related to the curriculum that should be taught, the language of instruction, learning providers, and certification (Brun and Shuayb 2020).

Furthermore, while education is expected to help children reach their aspirations, the legal status of refugees in countries offering short-term residence deters their ability to access the labor market and socially integrate. On the other hand, a long-term education is based on a developmental framework that speaks to the present and future. Hence, countries that offer refugees long-term legal residence provide children with training programs upon arrival. This enables them to follow the host country's national curricula and understand its language of instruction. Moreover, the long-term education paradigm recognizes refugee children's right to education as citizens and full members of society, which later empowers them to access employment and become socially integrated.

While the long-term paradigm carries appealing provisions that promote social and educational integration, we cannot overlook the fact that the short-term paradigm might also carry some advantages for refugee education. For instance, while school segregation in the short-term settlement negatively affects refugee children's social integration (Pugh et al. 2012), in some cases, it allows for special education provisions that are only feasible in segregated classrooms (Shuayb et al. 2022). One example would be the temporary education centers (TECs) in Turkey, where segregated Syrian refugee students learn using their home country curriculum by Syrian teachers and using their native language (Crul et al. 2019). Such provisions could later facilitate refugee children's integration into mainstream schooling, especially when accompanied by language and preparatory classes. Therefore, the flexibility of special provisions in segregated schooling can help refugee children continue their education upon arriving in the host country and prepare for their transition

to mainstream education. On the other hand, rapid school integration under the long-term paradigm improves social integration (Brun and Shuayb 2020); this often happens at the expense of language and learning difficulties, which in most cases diminish following the early stages of enrolment (Shuayb et al. 2022).

This motivates investigating how education policies and interventions might affect the absorption of refugees in host countries' societies, knowing that most countries are not prepared to absorb refugees when they come *en masse* in a short period.

## 3. The Multidimensionality of Refugee Integration

Following the influx of refugee children to their destination country, policymakers emphasize providing refugee children with access to education and give less attention to policies that respond to their needs that promote social integration in the medium- and long term (Pastoor 2016). This overlooks the fact that schools are not the only entities responsible for refugee students and that a comprehensive approach that includes different cultural, health, employment, housing, and welfare factors is essential for successfully integrating refugee children (Cerna 2019).

Refugee integration is a complex and gradual process requiring considering all social, economic, political, and cultural dimensions for their successful integration as members of the host society. Henceforth, one should consider all micro, meso, and macro-level factors to comprehensively investigate refugee children's integration determinants. The micro-level is concerned with individual factors that are important determinants of refugee students' social integration, knowing that refugee children's educational and social integration depends on a variety of individual and family factors such as race, ethnic background, socioeconomic status, and level or type of education of the parents (Cerna 2019). On the other hand, the meso level is concerned with school-level factors and in-class practices often shaped by policies and provisions at the macro level. Therefore, the meso level comprises factors such as type of schooling (integrated vs. segregated), the language of instruction, curriculum, training programs, and in-class practices and activities. The importance of school for refugee children lies in its significant role in providing a welcoming and respectful environment, which contributes to their successful social inclusion (Keddie 2010). The importance of macro-level factors stems from their effect on other direct micro- and meso-level determinants of integration. For instance, the literature widely discusses the effect of citizenship (macro-level policy) on living conditions (micro-level factors), whereby studies consistently claim that it will help create better living standards among naturalized refugees (Hutcheson and Jeffers 2012).

To a great extent, the design and implementation of macro-level policies are influenced by the type of legal status the host country offers to its refugees. Different types of legal status provide refugees with distinct educational paradigms that define and dictate their rights, level of education, and degree of social integration. For instance, countries offering short-term residence, mainly in the global south, are more likely to be concerned with the immediate humanitarian response, which lacks long-term educational and social integration goals. Other countries in the global north are more inclined to provide refugees with medium- to long-term residence, whereby refugees are fully integrated into all domains, such as education, employment, and culture, and may be granted full citizenship rights.

Integrating refugee children into the host country community is important for their education, which, if accompanied by other routes for social, cultural, and economic inclusion, determines their successful integration into the labor market and overall well-being (Cerna 2019). There is a scarcity of studies that examine the role of macro factors such as type of legal status and education paradigm (emergency vs. long-term) on factors on the meso (school and class practices) and micro (individual, household, and parental characteristics). Besides, we rarely see studies investigating refugee children's social integration in the global south and north. Therefore, the lack of research that encompasses all micro, meso, and macro factors in the global north and south has limited our understanding of what factors determine refugee children's social integration.

This paper investigates the determinants of refugee social integration and how different types of legal settlements shape refugee children's social integration. Henceforth, we selected three countries—Lebanon, Turkey, and Australia—which offer refugees different types of legal status. This paper attempts to address multiple gaps and, in doing so, makes important contributions to the existing literature. First, this study simultaneously investigates the micro, meso, and macro factors that shape refugee children's social integration. Second, this study's comparative approach encompasses both the global south and the global north. Third, this study **reveals the** implications of different education paradigms offered under different types of legal status.

## 4. Country Overview

Our data is collected from Lebanon, Turkey, and Australia because they all received Syrian refugees around the same time following the Syrian crisis in 2011. In addition, they offer refugees differ—ent types of legal status and have different institutional arrangements for integrating refugee children into education and society, which allows us to investigate the impact of legal status on their social integration.

In 2018, Lebanon was still considered an upper middle-income country with a gross domestic product per capita (GDPC) of over 8000 $ (World Bank 2018). Lebanon hosts the highest concentration of refugees per capita, with over 1.5 million Syrian refugees (VASyR 2019). However, Lebanon is not a signatory of the 1951 Convention Relating to the Status of Refugees nor its 1967 Protocol (UNHCR 2010). Lebanon offers Syrian refugees short-term residence and treats them as temporary visitors. As such, permanent residency, economic integration, socio-political integration, naturalization, and other longer-term solutions have been actively avoided, and interventions have been limited to temporary approaches. In collaboration with UN agencies, the Lebanese government designed the Reaching All Children with Education (RACE) strategy to provide all Syrian refugee students residing in Lebanon with education (Their World 2015). Both Race I (2014) and RACE II (2016) emphasized refugees' enrolment rates rather than the quality of education. RACE I's plan was temporary and assumed that repatriation would occur. Hence, it created educational challenges related to curriculum, the language of instruction, school segregation, certification, and official exams. With RACE I, refugee students learned using an outdated, rigid curriculum that was developed in 1997. They faced language difficulties learning math and science subjects in foreign languages and were segregated in public school's afternoon shift. RACE II in 2016 claims to implement development plans that would strengthen the Lebanese public education sector, improve the quality of education for all vulnerable children in Lebanon, and revise the national curriculum. However, very little was done, and in practice, these plans seemed to target nationals rather than refugees. While the focus on refugees remains on increasing enrolment and retention rates (Brun and Shuayb 2020), yet over 40% of school-aged refugee children in Lebanon have never been enrolled in education (NRC 2020).

According to the World Bank (2018), Turkey is also an upper-middle income with a GDPC of over 9000 $. Unlike Lebanon, Turkey is a signatory of the 1951 Refugee Convention and the 1967 Protocol (UNHCR 2019) and hosts the highest number (3.6 million) of Syrian refugees in the world. Although refugees in Turkey were offered short-term residence, the "temporary protection status" opened paths for permanent residency in 2014 and paths for citizenship in 2016. Henceforth, Turkey is considered a medium-term residency country, and its educational paradigm can be seen as a fusion of the EIE and long-term educational paradigm. Almost half of the refugee students were integrated into Turkish public schools, with the remaining half enrolled in temporary education centers (TEC). TECs were established to provide education to refugee children residing in and outside camps. These centers started as private initiatives that offered low fees, while others received financial support from the Turkish government and were free of charge. TECs taught Syrian refugee children using an almost identical curriculum to schools in Syria. Syrian teachers taught students using the Arabic language, yet they received some hours of Turkish

language lessons to prepare them to enroll in Turkish public schools. In 2016, the Turkish Ministry of Education announced that it would follow a new plan to integrate Syrian refugee children into mainstream education by enrolling all refugees in Turkish public schools within five years. Besides, Turkish public schools offered Syrian refugee children counseling facilities to help students facing challenges related to language, motivation, and learning qualifications. Despite all efforts to integrate refugee children and increase enrolment rates, one-third of school-aged children remain out of school in Turkey.

Australia is considered a high-income country with a GDPC of over 57,000 $ (World Bank 2018). Australia is also a signatory of the 1951 Refugee Convention and the 1967 Protocol (UNHCR 2019). Syrian refugees who applied through the Offshore Refugee and Humanitarian Settlement Scheme to Australia are offered permanent residence (Department of Home Affairs 2020). However, unlike the millions received by Lebanon and Turkey, Australia handpicked only 12,000 through their resettlement program. The Department of Social Services (DSS) ensures the development of policies and services that respond to refugees' needs in Australia. Additionally, the DSS is responsible for providing them with settlement services through the Humanitarian Settlement Program (HSP), which focuses on enhancing the social integration of refugees and assisting the transition to a self-reliant life in their settlement communities. The Australian Education Act 2013 indicates that schooling is compulsory for primary and secondary school years. The absence of a uniform government policy allows each state to design its own set of policies, programs, and activities such as the "New Arrivals Program". These programs provided all school-aged Syrian refugee children with the necessary education and language support to learn the Australian curriculum and become fully integrated into public schools. Furthermore, several non-government institutions and community programs and organizations have worked, either as independent entities or in partnership with each other and with the government, to support refugee settlement needs related to basic needs and services, health, employment, and education. These policies and provisions reflect Austria's long-term approach to a refugee-inclusive educational and social system.

This brief comparison of the three countries shows how the adopted education paradigm is a product of the legal status the host country offers to its refugees. Henceforth, this paper investigates the micro, meso, and macro factors that determine refugee students' social integration and looks at the implications of different education paradigms offered under different types of legal status (short-term, medium-term, and long-term). We propose the following hypothesis and research question about the nexus between refugee children's social integration:

**H1.** *Micro-, meso-, and macro-level factors are all crucial determinants of refugee children's social integration.*

R1. How the type of legal settlement and its corresponding education paradigm shape refugee children's social integration?

### 5. Data and Methodology

To answer the above research question and unpack how different education paradigms shape refugee students' social integration, we collected and analysed data at the macro, meso, and micro levels. The macro-level allows us to examine the policy framework. At the same time, the meso-level captures the schooling experiences, and the micro-level allows us to account for the individual characteristics of children. The study adopts a quantitative research methodology comprising a survey of grade 7, 8, and 9 students that include over 200 variables. The survey is part of a study titled: "Towards An Inclusive Education for Refugees: A Comparative Longitudinal Study", funded by The Spencer Foundation. Student surveys were conducted face-to-face by our researchers, who visited the selected schools and randomly chose a sample from each school. Data collection was followed by data entry, cleaning, and analysis using STATA. Further, all researchers have a certificate from the Collaborative Institutional Training Initiative, a research ethics and compliance

training program. As participants were below 18, their legal guardian consent was sought first. The names of participants were anonymized on the STATA file. Finally, the researchers would always ensure that the survey is carried out in conditions that allow privacy while at the same time ensuring the safety of both the participant and the researcher. Convenience sampling was used for this study. In Lebanon, this approach was necessary because of a lack of information regarding the target populations (Lebanese and Syrian students). Convenience sampling was also necessary because access to schools was determined by the Lebanese MEHE, which provided us with a list of public schools that included a large number of Syrian students. To reduce sampling bias, we collected data from all eight districts covering urban and rural areas. Similarly, in Turkey and Australia, convenience sampling was used with a focus on districts/states hosting a high refugee population.

In Lebanon, our survey covered 247 refugee students attending public schools only (morning and afternoon shifts) as it is currently the main provider of education for the vast majority of Syrian children attending school. However, as some students in Turkey were still enrolled in temporary schools run by Syrian community-based organizations, half of our student sample in Turkey was selected from temporary centers for a total of 710 refugee students. In Australia, we covered 341 refugee students attending state, catholic and independent schools where most Syrian are admitted across nine different districts where most new refugee families are resettled. In Lebanon, we covered all eight districts. In Turkey, we focused on two districts with the highest concentration of refugees: Istanbul and Gaziantep.

### 5.1. Variables

We start with defining our independent variable. In the survey, refugee children were asked to answer the following question, 'Do you feel welcome in your current country of residence?' Thus, the dependent variable is defined as a dummy variable that takes the value of one if the respondent is feeling welcome in the host country and zero otherwise. It is worth noting that the reference group is the category of students who are not feeling welcome in the host country. This variable serves as an accurate proxy of social integration since the more refugees feel welcome in the host country, the more attached they become to their new country of residence; thus, they become more willing to follow social norms and engage in social life (Laurentsyeva and Venturini 2017). Other studies (Constant et al. 2013) also rely on self-identification proxy variables to capture the general idea of social integration.

The independent variables are classified into four categories, individual and household factors, parental factors, school factors, and country of residence. Starting with individual and household characteristics, Gender is a dummy variable equal to 1 if the individual is a male and equal to 0 if female. Religion is also a dummy variable equal to 1 if the individual is a Muslim and 0 otherwise. Struggle to Pay Bills is a categorical variable equal to (1) if the student never struggles to pay bills, equal to (2) if the student sometimes struggles to pay bills, and equal to (3) if the student never struggles to pay bills, with category (1) taken as the reference group for all categorical variables. Type of Dwelling is a dummy variable equal to 1 if the student lives in a private apartment or a private house and 0 otherwise. Neighbors Mostly Displaced People is defined as a categorical variable equal to (1) if the student indicates it is not true that their neighbors are mostly displaced people, (2) if the student indicates it is somewhat true that their neighbors are mostly displaced people, and (3) if the student indicates it is true that their neighbors are mostly displaced people. We define Home-Country Area of Residence as a dummy variable equal to one if the student reported living in a city before moving to the host country, and 0 if the student reported living in a village before moving to the host country.

Moving to parental factors, we define Father's Education as a dummy variable equal to 1 if the respondent's father completed post-secondary education and equal to 0 otherwise. Similarly, we define Mother's Education as a dummy variable equal to 1 if the respondent's mother completed post-secondary education and equal to 0 otherwise. Father's Employment Status also serves as a dummy variable equal to 1 if the respondent's father is employed and 0 otherwise. Similarly, we define Mother's Employment Status as a dummy variable equal to 1 if the respondent's mother is employed and 0 if unemployed.

The third set of explanatory variables provides information on school factors such as school type, hours of language received by students per week, teachers' friendliness, and opportunities offered by the schooling system. We define Type of Schooling as a dummy variable equal to 1 if the refugee student is enrolled in a segregated schooling system such as afternoon shift in Lebanon or temporary education center (TECs) in the case of Turkey, and equal to 0 if the refugee student is enrolled in an integrated schooling system such as the morning shift in Lebanon, public school in Turkey, or any state school in Australia. Therefore, the entire Australian sample receives a zero for this variable, knowing that all refugee students in Australia are integrated into public schools. Hours of Language is a categorical variable equal to (1) if the student reported receiving five or fewer hours of language per week, (2) if the student reported receiving between 6 and 12 h of language per week, and (3) if the student reported receiving more than twelve hours of language education per week. Friendly Teachers is a categorical variable that is equal to (1) if teachers are always friendly, (2) if teachers are sometimes friendly, and (3) if teachers are never friendly. School System Offers Equal Opportunities is also a categorical variable equal to (1) if the student agrees their school system offers equal opportunities, (2) if the student reported "neutral" that their school system offers equal opportunities, and (3) if the student disagrees that their school system offers equal opportunities.

Finally, the last set of explanatory variables provides information about the student's current country of residence. We define Turkey as a dummy variable equal to 1 if the student currently resides in Turkey and 0 otherwise. Similarly, Lebanon is a dummy variable equal to 1 if the student resides in Lebanon and 0 otherwise. We included both country dummies in the model while treating Australia as a reference group.

Furthermore, we tested for multicollinearity problems that may lead to an increase in the variance of the regression coefficients, making our statistical significance inaccurate. This was done using one of the most common diagnostic tests, the Variance Inflator Factor (VIF), where a value of 10 or more for the VIF is considered problematic (Hair 2009). We find a mean value of 2, indicating the absence of a multicollinearity problem between our independent variables, with highest explanatory variable test value being 5.25.

*5.2. Summary Statistics*

Table 1 provides summary statistics for our explanatory variables. The table shows that around three-quarters of our sampled refugee students indicate feeling welcome in their country of residence. Furthermore, around 46% of our sampled refugee students are males, and around 92% are Muslims. Besides, around 31% reported that they always struggle to pay their bills, and around 85% are residing in a private house or private apartment. Almost 40% of our sampled refugee students reported that they reside in a neighborhood with mostly displaced people, while three-quarters of our sampled respondents reported they lived in a city before moving to their current country of residence.

**Table 1.** Summary Statistics of Variables.

|  | Number of Observations | Mean | Standard Deviation |
|---|---|---|---|
| **Dependent Variable** | | | |
| Feeling Welcome | 747 | 0.760 | 0.427 |
| **Independent variables** | | | |
| **Individual & Household Factors** | | | |
| Gender | 1290 | 0.455 | 0.498 |
| Religion: Muslim | 1299 | 0.923 | 0.267 |
| Struggle to Pay Bills: Sometimes | 1270 | 0.301 | 0.459 |
| Struggle to Pay Bills: Always | 1270 | 0.313 | 0.464 |
| Type of Dwelling: Private House/Apartment | 1299 | 0.853 | 0.354 |
| Neighbors Mostly Displaced People: Somewhat True | 1229 | 0.237 | 0.425 |
| Neighbors Mostly Displaced People: True | 1229 | 0.397 | 0.489 |
| Home-Country Area of Residence: City | 1242 | 0.750 | 0.433 |
| **Parental Factors** | | | |
| Father's Education: Post-secondary | 1278 | 0.322 | 0.468 |
| Mother's Education: Post-secondary | 1275 | 0.205 | 0.404 |
| Father's Employment Status: Employed | 1156 | 0.744 | 0.437 |
| Mother's Employment Status: Employed | 1241 | 0.135 | 0.342 |
| **School Factors** | | | |
| Type of Schooling: Segregated | 1299 | 0.426 | 0.495 |
| Hours of Language: 6 to 12 h | 1299 | 0.462 | 0.499 |
| Hours of Language: Over 12 h | 1299 | 0.259 | 0.438 |
| Friendly Teachers: Sometimes | 1248 | 0.151 | 0.358 |
| Friendly Teachers: Never | 1248 | 0.077 | 0.267 |
| School System Offers Equal Opportunities: Neutral | 788 | 0.156 | 0.363 |
| School System Offers Equal Opportunities: Disagree | 788 | 0.058 | 0.235 |
| **Country of Residence** | | | |
| Lebanon | 1299 | 0.190 | 0.393 |
| Turkey | 1299 | 0.547 | 0.498 |

We also observe that around one-third of our respondents reported that their fathers completed post-secondary education, while only around 20% reported that their mothers completed post-secondary education. Additionally, almost three-quarters reported that their fathers are employed, while only 14% reported that their mothers are employed.

The statistics reveal that approximately 43% of our sampled refugee students are enrolled in segregated schooling systems. Furthermore, around 46% reported receiving between 6 to 12 h of language classes per week, while only 26% of students reported receiving over 12 h of language classes per week. Besides, 8% of refugee students reported that their teacher is never friendly to them, and 6% do not agree that the school system offers students equal opportunities. Finally, 19% of our sampled refugee students are located in Lebanon, around 55% are located in Turkey and the remaining 26% reside in Australia.

*5.3. Limitations*

The main limitation of the study is the use of convenience sampling. Convenience sampling does not allow us to make generalizations about our target populations. As such, the quantitative results reported here are indicative of the phenomenon of interest and offer insights into refugee children's social integration. Another limitation of this study, which is common in comparative studies, is our inability to control for unobserved country characteristics (e.g., culture, racism) and pre-settlement conditions that might also affect

student social integration. Depending on the importance of these unobserved factors, our results might overestimate or underestimate the relationship between some of our chosen explanatory variables and social integration due to omitted variable biased.

*5.4. Empirical Model*

The primary objective of this study is to examine the determinants of refugee children's social integration and how different types of legal settlements shape refugee children' social integration. Our dependent variable, "Feeling Welcome", is a dummy variable; henceforth, we run the below regression using a probit[1] model in order to examine the determinants of refugee students' social integration:

$$\Pr(FW_i = 1) = \Phi(\beta_0 + \beta_1 IH_i + \beta_2 PF_i + \beta_3 SF_i + \beta_4 R_i)$$

We use the following probit observation rule:

$$FW = \begin{cases} 1 \text{ if the student is feeling welcome} \\ 0 \text{ if the student is not feeling welcome} \end{cases}$$

where $FW_i$ is our dummy dependent variable based on the following survey question: "*Do you feel welcome in your current country of residence?*" $IH_i$ is the vector of variables representing individual and household factors. $PF_i$ is the vector of variables representing parental factors. $SF_i$ is a vector of variables representing school factors, and $R_i$ is a vector representing the student's country of residence. $\Phi$ depicts the cumulative standard normal distribution function. Finally, $\beta_0$, $\beta_1$, $\beta_2$, $\beta_3$, and $\beta_4$ are vectors of individual parameters to be estimated. Probit is derived from a standard normal distribution[2] (a nonlinear function). Therefore, we apply calculus to observe our explanatory variables' effect on the probability of feeling welcome. These influences are known as marginal effects and are presented in Table 2, Column 4.

Table 2 presents a number of specifications of the model above. The first specification includes all individual, household, and parental variables (micro-level factors). We then add country fixed ($R_i$) dummies (macro-level factors) to the previously mentioned specification. The third specification adds to the first specification school-related variables (meso-level factors). Finally, the fourth specification includes all the control variables and controls for all micro, meso, and macro factors simultaneously. These specifications allow us to observe how the significance of micro and meso factors change after controlling for macro factors (country fixed effect dummies) and check how the significance of those macro factors changes as we gradually control for micro and meso factors.

**Table 2.** Determinants of Feeling Welcome (Probit Model).

| | (1) | (2) | (3) | **(4)** | **(4′)** |
|---|---|---|---|---|---|
| Individual & Household Factors | Coefficients | Coefficients | Coefficients | **Coefficients** | **Marginal Effects** |
| Gender | 0.050 | 0.030 | 0.032 | 0.025 | 0.007 |
| | (0.121) | (0.122) | (0.131) | (0.132) | (0.036) |
| Religion: Muslim | −0.351 | −0.188 | −0.277 | −0.278 | −0.077 |
| | (0.231) | (0.242) | (0.246) | (0.254) | (0.071) |
| Struggle to Pay Bills: Sometimes | −0.192 | −0.212 | −0.142 | −0.140 | −0.035 |
| | (0.158) | (0.160) | (0.170) | (0.171) | (0.042) |
| Struggle to Pay Bills: Always | −0.561 *** | −0.491 *** | −0.386 ** | −0.410 ** | −0.117 ** |
| | (0.155) | (0.162) | (0.172) | (0.174) | (0.048) |
| Type of Dwelling: Private House/Apartment | 0.281 * | 0.446 ** | 0.394 ** | 0.387 ** | 0.109 ** |
| | (0.168) | (0.181) | (0.182) | (0.188) | (0.052) |

**Table 2.** *Cont.*

| Individual & Household Factors | (1) Coefficients | (2) Coefficients | (3) Coefficients | (4) Coefficients | (4′) Marginal Effects |
|---|---|---|---|---|---|
| Neighbors Mostly Displaced People: Somewhat True | 0.174 | 0.095 | 0.182 | 0.199 | 0.044 |
| | (0.178) | (0.181) | (0.190) | (0.193) | (0.043) |
| Neighbors Mostly Displaced People: True | −0.369 ** | −0.430 *** | −0.406 ** | −0.382 ** | −0.112 ** |
| | (0.151) | (0.154) | (0.164) | (0.167) | (0.047) |
| Home-Country Area of Residence: City | 0.331 *** | 0.289 ** | 0.328 ** | 0.354 ** | 0.099 ** |
| | (0.128) | (0.131) | (0.138) | (0.140) | (0.039) |
| Parental Factors Father's Education: Post-secondary | 0.299 * | 0.238 | 0.220 | 0.252 | 0.070 |
| | (0.158) | (0.162) | (0.169) | (0.170) | (0.047) |
| Mother's Education: Post-secondary | −0.031 | −0.063 | −0.023 | −0.020 | −0.005 |
| | (0.207) | (0.208) | (0.222) | (0.223) | (0.062) |
| Father's Employment Status: Employed | 0.237 * | 0.464 *** | 0.504 *** | 0.480 *** | 0.135 *** |
| | (0.131) | (0.156) | (0.151) | (0.162) | (0.045) |
| Mother's Employment Status: Employed | −0.017 | −0.003 | 0.100 | 0.096 | 0.027 |
| | (0.193) | (0.197) | (0.214) | (0.215) | (0.060) |
| School Factors Type of Schooling: Segregated | | | −0.423 ** | −0.634 ** | −0.178 ** |
| | | | (0.175) | (0.291) | (0.081) |
| School System Offers Equal Opportunities: Neutral | | | −0.852 *** | −0.870 *** | −0.290 *** |
| | | | (0.166) | (0.169) | (0.061) |
| School System Offers Equal Opportunities: Disagree | | | −0.469 * | −0.447 * | −0.132 |
| | | | (0.251) | (0.254) | (0.085) |
| Friendly Teachers: Sometimes | | | −0.190 | −0.208 | −0.059 |
| | | | (0.172) | (0.173) | (0.052) |
| Friendly Teachers: Never | | | −0.859 *** | −0.858 *** | −0.297 *** |
| | | | (0.274) | (0.274) | (0.107) |
| Hours of Language: 6 to 12 h | | | 0.142 | 0.124 | 0.039 |
| | | | (0.158) | (0.165) | (0.053) |
| Hours of Language: Over 12 h | | | 0.351 * | 0.761 ** | 0.189 ** |
| | | | (0.209) | (0.374) | (0.079) |
| Country of Residence Lebanon | | −0.540 *** | | 0.278 | 0.077 |
| | | (0.197) | | (0.339) | (0.095) |
| Turkey | | −0.407 ** | | −0.204 | −0.057 |
| | | (0.204) | | (0.344) | (0.096) |
| Observations | 599 | 599 | 587 | 587 | 587 |

*Notes:* Standard errors in parentheses. Statistical significance *** $p < 0.01$, ** $p < 0.05$, * $p < 0.1$.

## 6. Empirical Findings

This section presents the determinants of refugee students' social integration by looking at micro-, meso-, and macro-level factors based on our quantitative analysis. Micro-level factors are used to assess the impact of individual, household, and parental characteristics on refugee students' social integration. In contrast, meso-level factors are concerned with the impact of different educational provisions and school practices on social integration. Besides, macro-level factors are used to reveal if social integration varies due to the differ-

ence in legal status offered by Lebanon, Turkey, and Australia. The quantitative analysis is based on the results presented in Table 2.

### 6.1. Micro Determinants of Refugee Students' Social Integration

Refugee children's educational and social integration depends on various individual factors such as race, ethnic background, socioeconomic status, and level or type of education (Cerna 2019). Our findings reveal that refugee students who always struggle to pay bills were 12% less likely to feel welcome than refugee students who never struggle to pay their bills, at a 5% significance level. Similarly, our results reveal that refugee students were 11% more likely to feel welcome when they live in a private house/apartment, at a 5% significance level. This is consistent with our a priori expectations since financial struggle and economic hardship faced by Syrian refugees act as a significant barrier to children's education (HRW 2015); this deprives children of access to schools, which is the primary site of social inclusion (Block et al. 2014). Living and interacting with nationals facilitates refugees' language acquisition, cultural familiarization, and social integration (FRA 2019). Results from this study support this claim as students residing in neighborhoods with mostly displaced people were 11% less likely to feel welcome than refugee students who share neighborhoods with the host community, at a 5% significance level.

Furthermore, refugee students who lived in a city prior to moving to their country of residence were 10% more likely to feel welcome than refugee students who moved from villages, at a 5% significance level. Similar results were shown in another study that involved refugee children, where social integration in the new country of residence was more challenging for children coming from agrarian societies compared to refugee children coming from cities (Segal and Maydas 2005). In addition, refugee students who reported that their fathers are working in their current country of residence were 14% more likely to feel welcome than refugee students who reported that their fathers are unemployed, at a 1% significance level. This is consistent with our a priori expectations since children coming from a working-class are more likely to socially integrate as they tend to participate in their parents' social interactions (Bernstein 1971). On the other hand, gender, religion, father's education, mother's education, and mother's employment were insignificant determinants of refugee students' social integration.

### 6.2. Meso Determinants of Refugee Students' Social Integration

The importance of school for refugee children lies in its significant role in providing a welcoming and respectful environment, which contributes to their successful social inclusion (Keddie 2010). According to Bridges and Walls (2018) educationally segregating refugee students was found to be one of the primary mechanisms that prevent children's social cohesion as it denies refugees the opportunity of learning the host country's language and culture from their national peers. Our findings support their claim, since refugee students enrolled in a segregated schooling system (afternoon shift in Lebanon or TECs in Turkey) were 18% less likely to feel welcome in their current country of residence than refugee students enrolled in an integrated schooling system, at a 5% significance level. Similarly, refugee students were less likely to feel welcome when their schooling system did not offer them equal opportunities, at a 10% significance level.

Moreover, when teachers were never friendly, students were 30% less likely to feel welcome compared to when teachers are always friendly, at a 1% significance level. This was in line with our expectations since schoolteachers and staff support at the school level is essential for providing a peaceful and friendly school environment that strengthens their inclusive classroom integration (Pugh et al. 2012). According to Cerna (2019), language is beneficial for refugee children's educational integration and essential to developing a sense of belonging within their new social environment. The results from this study support this finding since students who received more language classes per week were more likely to feel welcome in their current country of residence. This was mostly prominent when students received over 12 h of language education per week, whereby students receiving

more than 12 h were 19% more likely to feel welcome than students receiving between 0 and 6 h of language learning, at a 5% significance level.

The above sections presented our examination of the micro and meso levels where we quantitatively controlled for the country levels, i.e., the macro level in which the countries' legal status and education policies are manifested. The following section focuses on examining the impact of legal status and long-term versus emergency education responses on students' social segregation.

### 6.3. Macro Determinants of Refugee Students' Social Integration

Our country dummies depicting macro factors and legal status were only significant before controlling for meso-level factors, indicating that refugee students residing in Lebanon and Turkey were less likely to feel welcome than students living in Australia. However, these findings did not hold after controlling for meso-level factors whereby both country dummies (*Lebanon* and *Turkey)* became insignificant, indicating no significant difference in social integration between refugee students in Lebanon (short-term residency) and Turkey (medium-term residency) compared to Australia (long-term residency).

This is not consistent with our a priori expectations since, according to Crul et al. (2016), providing refugees with a path to naturalization is one of the best ways to integrate refugees fully into a new society. Similarly, Bloemraad (2006) argues that citizenship grants refugees rights and opportunities that strengthen their sense of belonging, while children growing up in a country that does not recognize them hinders their assimilation process (Saurer and Felfe 2014). Therefore, we expected to quantitatively observe significant differences in social integration between refugee children in Lebanon and Turkey compared to Australia, knowing that the latter offers a clear path for naturalization compared to the other two countries. One possible explanation concerning why no significant differences exist between these countries is that school factors in all three countries shape refugee social integration to a great extent. It statistically lessens the significance of the type of legal status (temporary vs. permanent) after being controlled for in the model. Specifically, the within-country differences because of the two parallel existing schooling arrangements (integrated vs. segregated) in Lebanon and Turkey are stronger than the between country differences, hence absorbing the significance of our country dummies. Therefore, this shows the importance of the meso-level factors that were omitted from the basic models and the significant association between our country dummies and the omitted variables. As we emphasized in the introduction, these meso and micro level variables are, however, also largely the result of national policies and the economic situation of a country at the macro level. Therefore, although our country dummies show no significant difference between countries, the country overview presented in the introduction, along with the following country comparison, is crucial to understand the importance of macro-level policies uniquely implemented by each country.

To reveal how different macro policies could shape other micro- and meso-level factors, we conducted a country comparison to observe variations in factors that are often indirectly driven by macro policies like school and housing policies. Table 3 presents the percentage relative frequency for some of our significant explanatory variables by country. The data reveals that refugee students in Lebanon struggled the most to pay their bills compared to refugees in Turkey and Australia. Additionally, received the least number of language hours per week, and as presented earlier, refugees in Lebanon were mostly segregated in afternoon shifts which hinders their social integration (Bridges and Walls 2018). In addition, a lower percentage of students in Lebanon reported that their teachers are friendly and that their school system offers equal opportunities compared to Australia. On the other hand, half of the Australian sample resides in neighborhoods with mostly displaced people, and only two-thirds reported living in private houses/apartments. To understand the lack of differences at the macro level between countries, we should take into account that potential negative factors at the meso and micro level that are the result of macro-level policies in Australia can level out other more positive factors on the macro level in Australia. This only

underscores the importance of investigating and scrutinize micro and meso-level factors in more detail.

**Table 3.** Country Comparison.

| Factor / Country | Lebanon | Turkey | Australia |
|---|---|---|---|
| Struggle to Pay Bills | | | |
| Always | 50.20% | 26.56% | 26.95% |
| Sometimes | 21.05% | 23.22% | 45.21% |
| Never | 28.74% | 50.22% | 27.84% |
| Type of Dwelling | | | |
| Independent House/Apartment | 92.31% | 92.96% | 64.52% |
| Other | 7.69% | 7.04% | 35.48% |
| Neighbors Mostly Displaced People | | | |
| TRUE | 48.78% | 27.83% | 56.53% |
| Somewhat True | 20.73% | 23.39% | 26.44% |
| Not True | 30.49% | 48.78% | 17.02% |
| Language Hours per Week | | | |
| 5 or less | 38.06% | 27.32% | 21.99% |
| 6 to 12 | 61.54% | 28.03% | 73.02% |
| over 12 | 0.40% | 44.65% | 4.99% |
| Friendly Teachers: *Sometimes* | | | |
| Always | 75.21% | 76.68% | 79.82% |
| Sometimes | 17.77% | 12.56% | 18.10% |
| Never | 7.02% | 10.76% | 2.08% |
| School System Offers Equal Opportunities | | | |
| Agree | 70.04% | 75.49% | 86.65% |
| Neutral | 22.27% | 14.71% | 11.28% |
| Disagree | 7.69% | 9.80% | 2.08% |

## 7. Conclusions

This study investigates the determinants of middle school refugee students' social integration in Lebanon, Turkey, and Australia. The three host countries offer refugees different types of legal status, allowing us to reveal if differences in refugees' social integration could be attributed to macro factors and the type of legal status offered. To fulfil this aim, our data collection covers a sample of 1298 middle school refugee students from all three countries.

Our findings reveal that micro factors significantly determine the level of refugee social integration. Refugee students who struggle to pay their bills and those living in neighborhoods with mostly displaced people are less likely to feel welcome in their current country of residence. On the other hand, refugee students living in a private house/apartment, students who lived in a city before moving to their current country of residence, and those whose fathers are currently employed are more likely to feel welcome.

Our quantitative findings reveal that meso-level factors are found to be the most significant determinants of refugee social integration. For instance, refugee students are less likely to feel welcome when enrolled in a segregated schooling system, when their teachers are never friendly, and when their school system does not offer equal opportunities. On the other hand, providing refugee students with intensive language classes increases their chances for successful social integration.

Macro-level factors that depict macro policies and legal status offered by the host country were only significant before controlling for meso-level factors. However, after controlling for meso-level factors in our model, macro-level factors became insignificant, indicating no important difference in refugees' social integration in Lebanon and Turkey compared to Australia. The statistical dominance of meso-level factors indicates that the within-country differences are stronger than the between-country differences, thus, absorbing our country dummies' significance. Therefore, our quantitative findings highlight the relevance of micro-and meso-level factors for refugee students' social integration. However,

it does not rule out the importance of macro-level factors, which according to the literature and our country comparison analysis, shape living and schooling conditions at the micro-and meso levels.

**Author Contributions:** Conceptualization, M.H. and M.C.; Data curation, M.H.; Formal analysis, M.H. and M.S.; Methodology, M.H.; Supervision, M.S. and M.C.; Writing—original draft, M.H. and M.C.; Writing—review and editing, M.H. and M.S. All authors have read and agreed to the published version of the manuscript.

**Funding:** This research was funded by The Spencer Foundation grant number 201800086.

**Institutional Review Board Statement:** This study was approved by the Institutional Review Board of the Lebanese American University LAU.STF.MS1.2018.R3.28/Apr/2021.

**Informed Consent Statement:** Informed consent was obtained from all subjects involved in the study.

**Data Availability Statement:** The data presented in this study are available on request from the corresponding author.

**Conflicts of Interest:** The authors declare no conflict of interest.

## Notes

[1] Probit model is a type of regression where the dependent variable can take only two values (Greene and Hensher 2010).

[2] The Standard Normal distribution, also known as the Z distribution, is one form of the Normal distribution in which the mean is equal to zero, and the variance is equal to 1 (Turner 2013).

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
