# Peer review of "Determinants of Refugee Children’s Social Integration: Evidence from Lebanon, Turkey, and Australia"

_socsci, doi:10.3390/socsci11120563_

Round 1

Reviewer 1 Report

The methodology shall be further developed to make it compatible with the statistical analyses used in other quantitative articles working with STATA and large datasets. The most important is to improve the hypotheses, and to make statistics acceptable to professional standards in the field os sociology.

Author Response

Thank you for taking the time to review our work. Kindly find attached the reply to your comments.

Reviewer 2 Report

This is a very interesting article. The authors would do well to better explain their selections including of case studies and the findings and discussions should occupy more space allowing deeper and greater analysis. My biggest concern is that the survey seems to have considered integration as a yes/no issue which seems very limiting. Allowing a scale might have benefited the researches nuance and depth. The exact role of education in the broader analysis could also be clarified further. Macro level issues beyond law and policy (e.g., racism and social exclusion) do not seem to be part of the analysis which is understandable but ought to be included as a limitation. Equally, the volume of migration (in terms of number, but also v GDP etc.) might also be an interesting angle. 

Author Response

(The authors gave the same response as above.)

Reviewer 3 Report

Dear authors, 

I like your research topic and find interesting. However, the manuscript is not well-structued and organized unfortunately. I have some recommendations about your paper. 

Authors cite too much the Cerna, 2019. They should vary their references. 

Some parts of statements are bold. Why? 

Your literature is not good organized. You sohould enough review the related literature. 

"To answer the above research question" I could not see where is it.

Why do you choose Convenience sampling? How about criterion sampling method?

"Segregated" is not true word. 

Where are your findings? I could not see it. 

There are some tables ans statistic, but it is not clear an understandable. 

There is no enough discussion in Results. 

Your references are not enough to explain your research topic. 

Author Response

(The authors gave the same response as above.)

Round 2

Reviewer 3 Report

Thanks for your revisions.